# Meta-Heuristic Model for Optimization of Production Layouts Based on Occupational Risk Assessment: Application to the Portuguese Wine Sector

António Agrela Freitas [1], Tânia Miranda Lima [1,2,*] and Pedro Dinis Gaspar [1,2]

1   Department of Electromechanical Engineering, University of Beira Interior, 6201-001 Covilhã, Portugal; antonio.freitas@ubi.pt (A.A.F.); dinis@ubi.pt (P.D.G.)
2   C-MAST—Centre for Mechanical and Aerospace Science and Technologies, 6201-001 Covilhã, Portugal
*   Correspondence: tmlima@ubi.pt

**Abstract:** A factory layout is a decisive factor in the improvement of production levels, efficiency, and even in the sustainability of a company. Regardless of the type of layout to be implemented, they are typically designed to optimize the work conditions and provide high performance, reducing production losses. The wine sector encompasses a wide diversity of possible configurations of production layouts, from one-level designs with separate infrastructures in several buildings or centralized single facilities, or even subdivided into different levels or floors. The general purpose is to maximize energy efficiency and process performance while minimizing costs. Thus, an optimization model based on the organization of productive layouts is proposed, using a methodology based on a genetic algorithm. The obtained results reveal that the optimization model for winery layouts was successfully applied, providing feasible solutions to improve the production processes' efficiency combined with the minimization of general and ergonomic risks.

**Keywords:** risk assessment; genetic algorithm; layout optimization; winery

## 1. Introduction

The production of table and liqueur wines includes several processes that are transversal to the production of red, rosé, and white table wines as well as some liqueur wines, namely, the generous wines of Porto and Madeira. It is noteworthy that these wines play a significant role in Portuguese wine exportation. Figure 1 shows the main unit operations related to the various winemaking processes for the different types of wines.

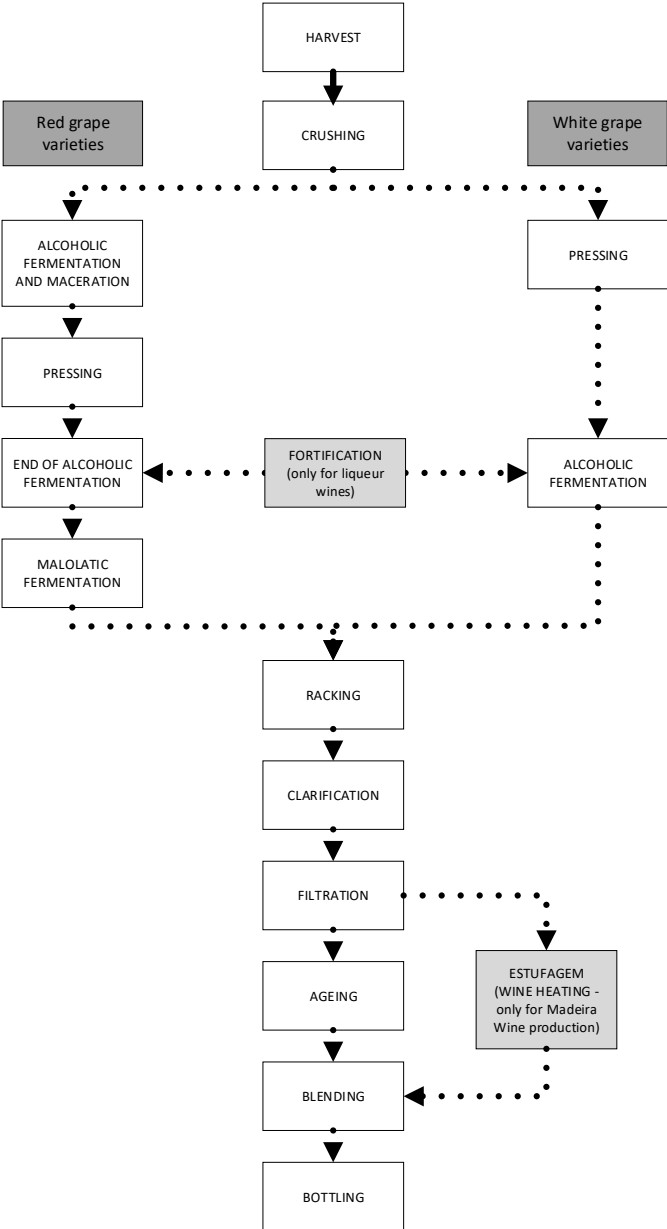

**Figure 1.** Schematic representation of the main unit operations related to the winemaking processes (Adapted from [1]).

Regarding the general occupational risks, according to Eurisko [2], the most critical accidents that have occurred in the beverage industry are related to explosions, followed, or not, by fire and deaths from suffocation. There are two common aspects in these situations: confined spaces and the presence of combustible and/or harmful particles or gases.

Eurisko, Anaya-Aguilar, Youakim, and Checchi [2–5] referred to the fact that the main ergonomic risks inherent in the wine production industry are:

- Musculoskeletal injuries (lumbar back);
- Exposure to vibrations;
- Physical fatigue;
- Visual fatigue.

The productive layout organization is one of the factors that contributes significantly to the reduction of occupational risks. According to Stephens and Meyers [6], planning manufacturing facilities is a multifaceted process, influenced by numerous factors and

variables, such as economics factors (i.e., tax incentives or geopolitical considerations), that are not always in harmony and sometimes can even have a contradictory impact on the decision-making process.

The design of manufacturing facilities involves the organization of the company's physical assets in order to promote the efficient use of resources such as human resources, materials, equipment, and energy. The facility design includes the factory location, the building design, the factory layout, and the materials handling systems [7]. The industrial design of facilities and the materials handling affect the company's productivity and profitability more than any other major corporate decision [8].

The layout is the visual presentation of the data and analyses performed during the planning process. The combination of accuracy, data credibility, and logical analysis of information can lead to obtaining a functional layout, or increasing efficiency in the case of existing layout optimization processes [6].

As stated by Moran [7], the design of an organization's manufacturing facility depends on nine very important steps:

- Gathering information;
- Establishing a time standard;
- The process design;
- The flow analysis;
- Analysis of activity relationships;
- Ergonomics and space requirements;
- Space requirements for auxiliary services (e.g., storage of raw materials);
- The handling of materials;
- Determination of equipment for handling materials.

In this research, based on the results of the risk assessments performed, an optimization model to improve production layouts is proposed to obtain feasible solutions that can integrate all of the operations that take place according to the winery's activity, both regarding the production processes operations and the minimization of general and ergonomic risks.

The layout of a factory is the decisive factor in terms of production levels, efficiency, and even the sustainability of the company. Regardless of the type of layout to be implemented, they are usually designed to optimize the working conditions and provide good performance, minimizing production losses [9]. Solving most assembly line layout problems applies decomposition algorithms, multi-stage algorithms, genetic algorithms, and other techniques [10].

In order to solve most layout problems, several techniques are often developed and applied, such as decomposition algorithms, multi-stage algorithms, genetic algorithms, and particle swarm optimization algorithms [10].

Regarding the optimization of wine production facilities, Torreggiani et al. [11] determined two possible layout solutions: an axial layout solution and a compact layout solution, which were obtained through the application of a meta-design methodology after the establishment of direct and indirect relationships between the various space units, whereas Gómez, Tascón, and Ayuga [12] proposed several types of layouts, which were obtained through the use of the systematic layout planning method, taking into account the continuity of the established production process as well as the criteria of product quality and hygiene, noise, smells, accessibility, hygiene and safety at work, and the difference between wet and dry areas.

Despite the feasible results provided by these methodologies, due to the inclusion of additional variables, such as environmental changes, they can become complex and time consuming. For this reason, heuristics or meta-heuristics are often used to obtain near optimal solutions, providing flexibility and the capacity for adaptation as the environment changes.

The most common computational intelligence algorithms to solve the problem of layout optimization are genetic algorithms (GAs), which are particularly suitable for solving complex optimization problems and, consequently, are effective for applications requiring

adaptive troubleshooting strategies [13]. The theoretical basis for GAs was proposed by John Holland in 1975 [14]. The GA is a model that reproduces some evolutionary biological theories in the resolution of optimization problems [15].

Numerous applications for GA have been proposed to solve the facility site layout problem. RazaviAlavi and AbouRizk [16] proposed an integrated simulation model based on a genetic algorithm to estimate the minimum layout cost, taking into account restrictions such as required distances for safety or accessibility between facilities.

To define the ideal design for a distribution system in a logistic distribution centre, Chen et al. [17] suggested a method based on a genetic algorithm that allows for the capacity maximization of a distribution system with multiple charging stations.

Liu et al. [18] applied a genetic algorithm to innovatively optimize offshore wind farm layouts under different seabed terrains, increasing the total energy output, while Krajčovič et al. [19] presented a specific genetic algorithm layout planner (GALP) that uses a genetic algorithm to optimize the spatial arrangement.

Said and El-Rayes [20] proposed and compared two global optimization models of dynamic site layout planning, where the first model utilized GAs, while the second model was based on an approximate dynamic programming.

Despite the fact that the occupational health and safety field has always followed revolutionary developments in the industry, reacting positively to technological progress and changes, leading to reliable and sustainable solutions, there is a lack of research work based on facility layout problems that include the occupational risk aspect [21].

Thus, this work aimed to contribute to filling this gap in the literature by proposing a different method based on a genetic algorithm to solve a facility layout problem considering the occupational risk component, allowing for the mitigation of occupational risks allied to a higher level of operational process efficiency.

Therefore, the objectives of this study are summarized as the following:

- Identify and quantify two categories of risks related to the wine sector:
    - General;
    - Ergonomic.
- Implement a method to optimize wine industry layouts with the view of occupational risk mitigation.

To achieve these objectives, this study integrated two methods of risk assessment based on:

- The William T. Fine method for general risk;
- Metabolic energy expenditure for ergonomic risks.

In a subsequent step, an approach based on a genetic algorithm provides the optimization of the layout through a spatial reorganization according to the distances between the different areas of the layout, taking into account the results of the risk assessment.

Both risk assessment methods allowed for the quantification and location of the different levels of occupational risk, making it possible to establish two different profiles, the results of which provided the layout optimization model.

On the other hand, the optimization model provided feasible solutions and contributed to the improvement of the production processes' efficiency combined with the mitigation of occupational risks.

## 2. Material and Methods

This work comprised two phases. The first involved the occupational risk assessment related to the wine industry sector, which includes the identification, evaluation, and quantification of occupational risks, whereas the second phase consisted of the optimization of production layouts based on the application of a genetic algorithm.

The first phase included two steps: a first step based on a general risk evaluation performed based on the semi-quantitative William T. Fine method, and a second step that involved an ergonomic risk assessment relying on metabolic energy expenditure. In the

first step, the William T. Fine method was applied by determining the values of the average danger level for each area, which was based on the risks and respective risk danger levels in the work carried out by Eurisko, Anaya-Aguilar, Youakim, and Checchi [2–5]. In the second step, the ergonomic risks' evaluation and quantification were obtained from the metabolic energy expenditure estimation that was based on the energy expenditure involved in the several tasks that integrate the production processes which, in turn, were obtained by comparison to several activities already measured and characterized in several studies by Ainsworth et al. [22–24].

Finally, the optimization of production layouts was based on the application of a genetic algorithm that included the results of the previous phase. The GA was chosen due to the fact of its versatility and embracing nature, providing a process with random effects, according to Sariff and Buniyamin [25], allowing for the exploration of several alternatives in a non-conditioned way.

*2.1. The Optimization Model*

The solution procedure of the applied model was based on a genetic algorithm that integrated mutations throughout the various iteration processes and through the action of a tournament selection operator.

Figure 2 shows the flow chart for the execution of a typical genetic algorithm. First, the type of variables and their coding for the problem in question should be defined. Then, the fitness function should be stipulated, and it often corresponds simply to the objective function to be optimized. In general, the fitness function can be any function that assigns a merit value relative to an individual. Genetic operators, such as crossover and mutation, are applied stochastically at each stage of the evolution process, so their probabilities of occurrence must be defined. Finally, convergence criteria should be provided [26].

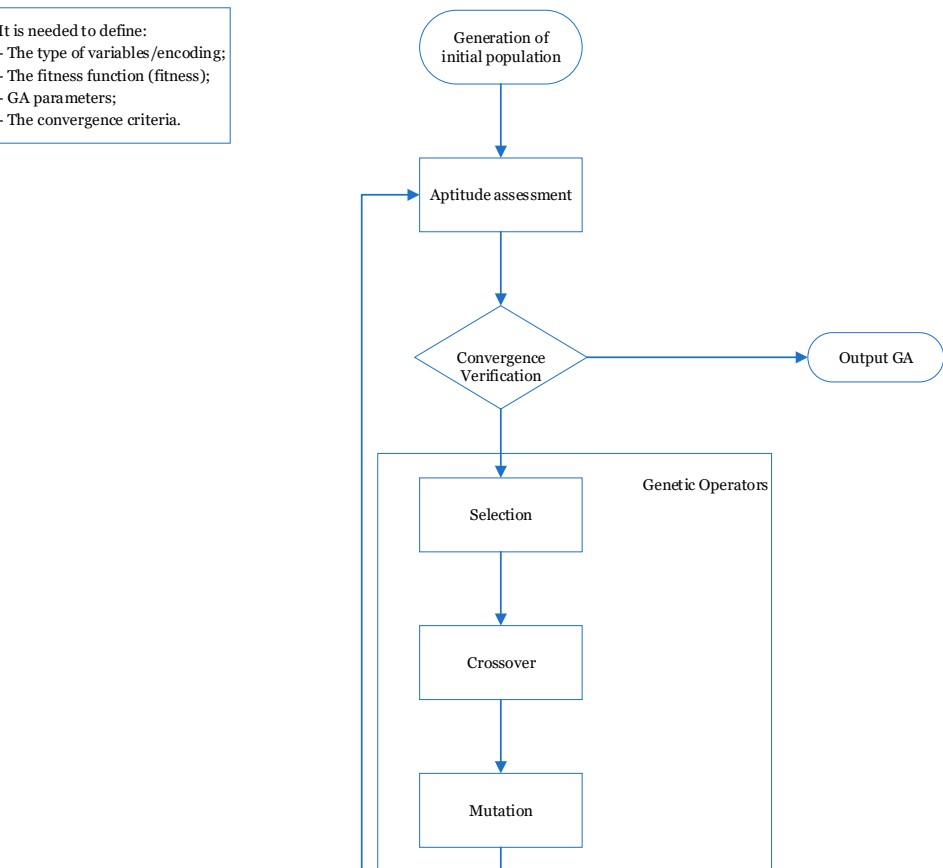

**Figure 2.** GA implementation flow chart [26].

The algorithm begins by defining the problem and its objective function, f(X), where X is a multidimensional vector of typical dimension d. The initial population is chosen randomly in the search space, and the members are encoded as a chromosome in the form of a string of alphabets. The operations of selection, crossover, and mutation are cyclically applied to the population until the termination criterion is achieved or the maximum number of iterations is reached. At the end of each iteration, the fitness values of the population of strings are calculated. The strings with higher fitness are selected for mating and reproduction. Finally, the string with the highest fitness value is considered the optimum solution to the problem.

The pseudocode for the genetic algorithm (Algorithm 1) is illustrated as follows [27].

---

**Algorithm 1:** The pseudocode for the genetic

---

**Initialization:**
   1. Select initial population of size $N$
   2. Define objective function $f(X)$
   3. Encode the population as chromosomes (bit strings) of length $L_C$
   4. Compute fitness values of the entire population
   5. Define termination condition, if any
   6. Choose maximum number of iterations MaxIter
   7. *iter* = 1
   8. **while** (*iter* $\leq$ *MaxIter*)
**Selection:**
   9. Select parents for reproduction
**Crossover:**
   10. Apply crossover on parents to produce offspring
**Mutation:**
   11. Apply mutation on selected chromosomes
   12. Compute the fitness values of the population
   13. Select members for the next generation based on fitness values
   14. **If** termination condition met exit, **else** continue
   15. *iter* = *iter* + 1
   16. **end while**

---

Thus, the model aims to enable the optimization of the layout of a productive area through spatial reorganization, considering the distances between the various areas in the layout represented by Euclidean space (abstract mathematical space) based on the minimization of costs [28], according to two distinct approaches, considering:

- The costs in terms of general risks, which were based on the William T. Fine semi-quantitative assessment method, being applied by determining the values of the average danger level for each area, which was based on the risks and respective risk danger levels in the work carried out by Eurisko, Anaya-Aguilar, Youakim, and Chec-chi [2–5]. The calculation of the values related to the average risk level was made using Equation (1):

$$RL = C \times E \times P \tag{1}$$

where *RL* corresponds to the risk level, *C* to the expected consequences, *E* to the time of exposure of the worker to the risk situation, and *P* to the probability of occurrence.

The risks and degrees of danger can then be classified into five levels: Acceptable, Moderate, Remarkable, High, and Very High (severe and imminent).

- The costs in terms of ergonomic risks based on the assessment of ergonomic risks related to the expenditure of metabolic energy that occurs during the execution of the various tasks within each work area. It is noteworthy that an estimation of metabolic energy values was made based on the comparison with several tasks already studied

by Ainsworth et al. [22–24] having been made an average for each work zone as shown in Table 1.

**Table 1.** Example of a comparative estimation of the metabolic energy that is normally spent in carrying out the task of bottling according to the tasks studied in several research works.

| Activity | Main Tasks | Comparative Tasks Studied by Ainsworth et al. [22–24] | Estimation of Metabolic Energy Expenditure | | | Level |
| --- | --- | --- | --- | --- | --- | --- |
| | | | MET | J/s | kcal/min | |
| Bottling | Operate/control filling system | Operating heavy, automated equipment, does not include driving vehicles/equipment | 2.5 | 216.3 | 3.1 | Light |
| | Cleaning/preparation of the filling system (manual handling of loads) | Transporting moderate loads upstairs, moving 11–22 kg boxes | 8 | 697.8 | 10 | Very Heavy |
| | Maintenance of the filling system (manual handling of loads) | Light/moderate standing work (e.g., assembly/repair of heavy parts, welding, wrapping parts, car repair, and box packing) | 3 | 265.2 | 3.8 | Light |
| | Access to the filling system from the top (unlevelled work—going up and down stairs) | Climbing the stairs | 7.5 | 655.9 | 9.4 | Heavy |
| | | Going down the stairs | | | | |
| | Moderate work in a standing position | | 3.5 | 307.0 | 4.4 | Light |
| | | **Average:** | **4.9** | **425.7** | **6.1** | **Moderate** |

The two approaches were applied with the purpose of verifying the effect of general risks on the reorganization of productive areas compared to the influence of ergonomic risks and their impact on the efficiency of functionally defined areas.

Regarding the number of work zones, this parameter was incorporated in proportion to the global area, which was determined based on the input parameters' width and length of the building as shown by Equation (2):

$$N_z = L_y \, L_x \qquad (2)$$

where $N_z$ is the number of zones to integrate in the plant, $L_y$ is the length of the facilities, and $L_x$ is the width of the building.

Thus, to determine the number of zones, the following procedure establishes the relationship between the initial variables and the initial parameters.

If the number of arguments is less than two:

```
Then
    P_size = 50 × 2;
    T_i = 2500;
    L = 3 ; W = 5;
    N_z = Round(3 × 5);
    Ld = randi(80, [N_z N_z]);
    Ld = [L_1  L_2  L_3  L_4  L_5  L_6  L_7  L_8  L_9  L_10  L_11  L_12  L_13  L_14  L_15];
    LD = Ld + Ld';
    Loads = LD;
else
    N_z = Round(L_y W_x);
end
```

where $P_{size}$ is the population size; $T_i$ the number of iterations; $L$ and $W$ are the length and width of the building, respectively; $N_z$ is the number of rectangular facilities of equal areas; $Ld$ is the load matrix, which contains the costs (data are shown in Appendix A) in terms of general occupational risks, in the first stage, or the costs in terms of ergonomic risks (data are shown in Appendix B) at a later stage. It should be noted that the equation, $Ld = randi(80, [N_z \ N_z]$, aims to randomly assign notional loads.

The genetic algorithm initiates from a primary set of random solutions, which constitutes the population. The size of the population, P ($P_{size}$), corresponds to the number of solutions generated, also known as chromosomes. Each solution is evaluated according to a predefined aptitude function [29]. Chromosomes evolve through successive iterations, known as generations, and each generation is represented by a new population, most often different from the previous one [30].

The new populations are not randomly generated because the chromosomes that are part of the current population are selected, mixed, and modified through genetic operators, such as selection, crossing, and mutation, to generate new chromosome sequences [31]. The new population, known as offspring, incorporates only the fittest chromosomes into the offspring, inhibiting the chromosomes that are less suitable. In this way, it is guaranteed that future generations will always be better than the previous ones. Thus, the new populations will be generated iteratively until the specific stop criterion is reached and, in this case, the maximum number of iterations will be reached [27].

It is necessary to establish a coding scheme to represent the parameters of the problem in the string of chromosomes. For a discrete representation of the problem, the entire factory is divided into N zones, and each zone corresponds to one location. These locations are characterized by two parameters: their spatial location and their assigned sequential position [32].

### 2.1.1. The Fitness Function

The fitness function corresponds to the objective function of the problem, which in this type of problem corresponds to the total cost of a given solution, represented by *Cost (S)* in Equation (3).

Considering the work of Gonçalves and Resende [33], with the necessary adaptations and inclusion of new specificities, the layout was defined by the coordinates of the centroid $(x_i, y_i)$, and the horizontal ($w_i$) and vertical ($l_i$) dimensions of each zone $i$ which, in turn, are sized according to the layout dimension and, consequently, depend on the number of areas, $N_z$.

Each solution corresponds to the location of $M$ areas in $R \ (\geq M)$ positions [33].

$$Cost \ (S) = \sum_i^M C_a(i, S(i)) + \sum_i^M \sum_j^M [C_g(i,j) \ D(i,j)] \tag{3}$$

$$C_a(S(i)) = Z \sum_i^M N_i \tag{4}$$

where the first term represents the sum of all costs, $C_a$ (units: m or J·m/s depending on the type of occupational risk) related to the shared zones between the productive areas, which are used for moving materials, products, and people. This term takes into account the general occupational risks (without units) or ergonomic risks (Joule) given by a factor $Z$ imputed to each common zone, $N_i$. The second term is the sum of all costs $C_g$ (units: m or J·m/s depending on the type of occupational risk) related to the occupational risk loads existing in each location area $S(i)$, whether in the case of general (without units) or ergonomic risks (Joule/second), taking into consideration the Euclidean distance between two zones $d(ij)$ (Equation (5)), which can be considered in meters ($m$) as a practical aspect. The fitness function aims to determine the minimum cost in consonance with the objective function.

$$D(i,j) = \sqrt{\sum_i^j (x_i - x_j)^2 + (y_i - y_j))^2} \, i, j \in M \tag{5}$$

The objective function (Equation (6)) minimizes the total cost, in terms of occupational risks, using the appropriate distance norm. Thus, the objective function is given by Equation (5):

$$Min\ Cost\ (S) = \sum_i^M C_a(i, S(i)) + \sum_i^M \sum_j^M [C_g(i, j)\ d(ij)] \tag{6}$$

The model uses the additional parameters $P_{i,j}, Q_{ij}$, which are binary variables used to model the non-overlapping constraints, where:

If $(P_{ij}, Q_{ij})$ is equal to (0,0), then the facility $i$ is forced to move to the right of $j$;

If $(P_{ij}, Q_{ij})$ is equal to (1,0), then the facility $i$ is forced to move to the left of $j$;

If $(P_{ij}, Q_{ij})$ is equal to (0,1), then the facility $i$ is forced to move to above of $j$;

If $(P_{ij}, Q_{ij})$ is equal to (1,0), then the facility $i$ is forced to move below of $j$.

$M_x\ and\ M_y$ are parameters that aim to define the upper bounds on the horizontal and vertical planes between any two facilities, respectively.

$d_{ij}^x\ and\ d_{ij}^y$ are variables that represent the distances between the facilities $i$ and $j$ along the $x$- and $y$-axes, respectively.

The objective function (Equation (6)) is subject to following constraints:

Facility constraint:

$$w_i\ l_i = A_i \qquad \forall i \in M \tag{7}$$

Non-overlapping constraints:

$$x_i - x_j + M_x(P_{ij} + Q_{ij}) \geq \frac{w_i + w_j}{2} \qquad \forall i, j \mid j > i \tag{8}$$

$$x_j - x_i + M_x(1 - P_{ij} + Q_{ij}) \geq \frac{w_i + w_j}{2} \qquad \forall i, j \mid j > i \tag{9}$$

$$y_j - y_i + M_y(1 + P_{ij} - Q_{ij}) \geq \frac{l_i + l_j}{2} \qquad \forall i, j \mid j > i \tag{10}$$

$$y_j - y_i + M_y(2 - P_{ij} - Q_{ij}) \geq \frac{l_i + l_j}{2} \qquad \forall i, j \mid j > i \tag{11}$$

Spatial constraints:

$$\frac{w_i}{2} \leq x_i \leq W - \frac{w_i}{2} + \Delta w_i \qquad \forall i \in M \tag{12}$$

$$\frac{l_i}{2} \leq x_i \leq W - \frac{l_i}{2} + \Delta l_i \qquad \forall i \in M \tag{13}$$

Distance constraints:

$$x_i - x_j \leq d_{ij}^x \qquad \forall i, j \mid j > i \tag{14}$$

$$x_j - x_i \leq d_{ij}^x \qquad \forall i, j \mid j > i \tag{15}$$

$$y_i - y_j \leq d_{ij}^y \qquad \forall i, j \mid j > i \tag{16}$$

$$y_j - y_i \leq d_{ij}^y \qquad \forall i, j \mid j > i \tag{17}$$

Domain constraints:

$$x_i, y_i, w_i, l_i, d_{ij}^x, d_{ij}^y \geq 0 \qquad \forall i \in M \tag{18}$$

$$P_{ij}, Q_{ij} \in \{0, 1\} \qquad \forall i, j \mid j > i \tag{19}$$

The constraint (7) defines the horizontal and vertical dimensions of the facility according to the area and maximum allowed ratio. The constraints (8) to (11) enforce the non-overlapping constraints by imposing the facilities to be separated horizontally and vertically. The constraints (12) and (13) force each zone to be within the horizontal and

vertical limits of the building space, respectively. The constraints (14) to (17) define the distances between all pairs of zones (*i,j*) according to the distance function. Finally, the constraints (18) and (19) define the variables' domains.

### 2.1.2. The Genetic Operators

There are three basic genetic operators that are used to generate a new population in each generation: selection, crossing, and mutation [27]. The selection corresponds to the process of sampling solutions of the current population. It is represented by a skewed selection process that is used to determine which solutions should be included in the new population. The method used in this implementation was selection by tournament [34]. The solution selection criterion, to integrate the next generation, is based on the fitness value of each solution, which is converted into a probability of being selected. The most appropriate solutions are more likely to be chosen compared to the least appropriate solutions [35].

The crossover operator is used to reproduce the chromosomes of the offspring by crossing between four parent chromosomes as in the below [32,36].

Parent 1 = |1 2 3 4 5 6 7 8 9 10 11 12|
Parents:
Parent 2 = |3 2 8 1 4 9 6 5 7 12 10 11|
After three random cut-off points:
Parent 1: |1 2 |3 4 5 6|7 8 9 |10 11 12|
Parents:
Parent 2: |3 2|8 1 4 9|6 5 7|12 10 11|

The genes are copied between the two cut-off points of the first parent. The remaining genes are copied from the second parent from the second cut-off point, excluding genes already transferred to the descendant and preserving order [36].

Offspring 1: |1 2|8 1 4 9|7 8 9|12 10 11|
Offspring 1: |3 2|3 4 5 6|6 5 7|10 11 12|

First, a random position (cut zone) is selected in both parents, which divides the information from the parent to be included in the descendant. This cutting position must ensure that there is at least one position occupied in the partial information of both parents, otherwise the offspring may lose zones in the process and generate invalid solutions. The offspring is conceived by including partial information from both parents, assigning the spatial location and sequential position information to the new positions [36]. At this stage, the same zone can be assigned to more than one location. However, since there can be no duplicate installations, a replacement must be made in advance. In other words, in these cases, the position of the first occurrence in the duplicate zones in the offspring, which corresponds to the partial information of the second parent, must be determined. This position is used to find a different zone in the first parent. If this new zone already exists in the offspring, the process is repeated until the chosen zone is not removed from the descendant. Otherwise, the new zone will be included in the descendant. If there is a duplicate zone that cannot be avoided (no substitute zone has been found), then the descendant is deleted and the relative is included in the offspring [32,36].

The mutation operator is used to introduce randomness into a solution, preventing it from getting stuck in a minimal place. The purpose of this mutation operator is to exchange only the status and equipment information between two positions chosen on a chromosome [37].

### 2.1.3. The Algorithm Execution

The algorithm used follows the following general steps [38]:

- Initialization: The number of zones is obtained according to the dimensions of the layout, based on the length × width = number of zones (3 × 5 = 15 zones). It is noteworthy that the risk matrix indicates the degree of average hazards per zone according to the risk assessment carried out. The distance matrix is based on the Euclidean distance between the zones, given by Equation (5);

- Random population generation to be submitted in permutations for the layout optimization;
- Minimum cost selection of the for all populations (attempts) keeping the best members and being made the respective graphic representation;
- Crossover of the elements of the population for a new tournament;
- Subdivision of the population into groups of four elements;
- Selection of the best of the four elements and substitution of the worst of the four elements of the subgroup population;
- Mutation of the best of the four elements (the winner) in each subgroup;
- Incorporation of the best of the four elements (the winner) and all mutations, however, affected in the population;

If the result of the iteration remains, it returns to running from the fourth step. Otherwise, the termination is final:

- Termination: based on the result of the iteration.

### 2.2. Application of the Optimization Model

Having as a starting point a project for a winery (Figure 3), according to the design of Öztürk [39], the winery layout is divided into two distinct levels or floors. The first level is composed of all the areas inherent to production, as well as other areas associated with the reception of customers, including a restaurant area, a sales shop, and areas for exhibitions and workshops, while the second level or floor is intended exclusively for the storage and staging of wines in stainless-steel vats or barrels.

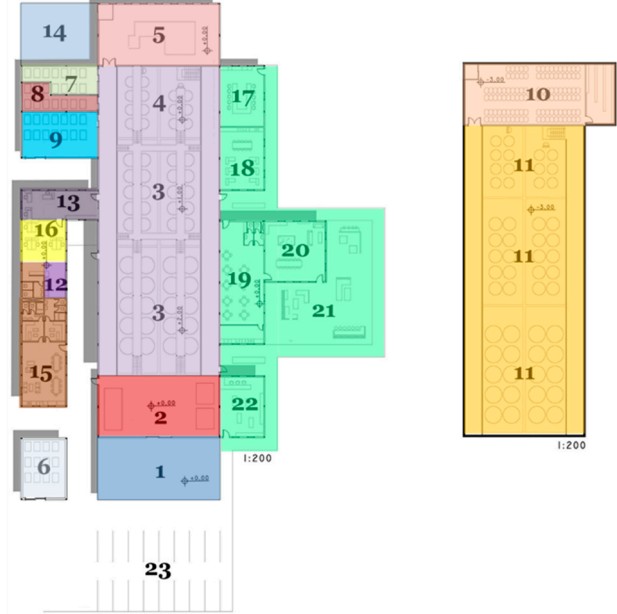

**Figure 3.** Example of a winery layout [39]. (1: Area of reception of the raw materials (i.e., grapes); 2: winemaking area; 3: fermentation area; 4: batches working area; 5: bottling area; 6: area of clarification/and stabilization; 7: warehouse for finished product; 8: warehouse for semi-detached product; 9: warehouse for packaging materials; 10: wine ageing in barrels area; 11: storage area/batch preparation—stainless-steel vats; 12: laboratory; 13: technical area (oenology); 14: shipping area; 15: cafeteria/social area for employees/sanitary facilities; 16: offices; 17: auditorium; 18: workshop area; 19: restaurant; 20: wine tasting room; 21: terrace (outdoor); 22: shopping area (public sales); 23: carpark).

Although there are numerous advantages in the distribution of work areas in terms of differentiated implementation quotas at various levels or floors, in the present study, we considered the location of all functional areas at the same level, a scenario that can be

found in most wineries in operation. The optimization model was applied to the work areas according to Table 2.

**Table 2.** Work zones considered in the application of the optimization model.

| | Areas |
|---|---|
| 1. | Reception of Raw Materials (i.e., Grapes) |
| 2. | Vinification |
| 3. | Fermentation |
| 4. | Clarification/Stabilization |
| 5. | Filtering |
| 6. | Wine Storage/Conservation in Stainless-Steel Vats |
| 7. | Wine Storage/Ageing in Wooden Barrels |
| 8. | Estufagem (Wine Heating)—only in Madeira Wine Production |
| 9. | Batches Preparation |
| 10. | Bottling |
| 11. | Packaging Materials Warehouse |
| 12. | Semi-Finished Product Warehouse |
| 13. | Finished Product Warehouse |
| 14. | Packaging |
| 15. | Expedition |

The laboratory areas and the technical area (oenology) were not considered in the optimization of the layout since they are auxiliary support divisions that are related to all productive areas. Additionally, the filtration and the blending areas, which are dissociated from the storage area of wines in stainless-steel vats, were considered, and the wine heating area was considered as well, which is specifically for the production of Madeira wine.

## 3. Optimization Results

For each evaluation, four scenarios were considered, as specified in Table 3, according to the production processes, with the inclusion and exclusion of the wine heating area, which is a specific operation for the production of Madeira wine, regarding the type of risk assessment and general and ergonomic risks.

**Table 3.** Established scenarios for the application of the optimization models, taking into account the assessment of general risks.

| Scenarios | Wine Production Profile | Types of Risk Evaluation | Objective and Optimization Model |
|---|---|---|---|
| 1 | Usual wine production including table wines, Port wines, and Madeira wines (with the wine heating process) | General risks | Production layout optimization Occupational risks minimization |
| 2 | | Ergonomic risks | |
| 3 | Usual wine production including table wines and Port wines. Exclusion of Madeira wine production (without the wine heating process) | General risks | |
| 4 | | Ergonomic risks | |

The optimization model based on a genetic algorithm, the objective of which is to reduce costs, included the average degree of hazard determined for each area of labour, taking into account the spatial location, which is determined according to the dimensions of the building (length and width), based on Euclidean geometry

The results obtained in the general and ergonomic risk assessments are synthesized in Table 4, wherein the general risk assessment was performed by applying the William T. Fine method (Appendix A) (scenarios 1 and 3), and the ergonomic risk assessment was performed through the metabolic energy expenditure estimation (Appendix B) (scenarios 2 and 4).

**Table 4.** Synthesis of the risk assessments applied to the four scenarios.

| Areas | Scenario 1 | Scenario 2 | | Scenario 3 | Scenario 4 | |
|---|---|---|---|---|---|---|
| | Danger Level | J/s | kcal/min | Danger Level | J/s | kcal/min |
| 1. Reception of Raw Materials (i.e., Grapes) | 467.1 | 474.5 | 6.8 | 467.1 | 474.5 | 6.8 |
| 2. Vinification | 574.6 | 474.5 | 6.8 | 574.6 | 474.5 | 6.8 |
| 3. Fermentation | 600.4 | 467.5 | 6.7 | 600.4 | 467.5 | 6.7 |
| 4. Clarification/Stabilization | 436.3 | 390.8 | 5.6 | 436.3 | 390.8 | 5.6 |
| 5. Filtering | 462.9 | 523.4 | 7.5 | 462.9 | 523.4 | 7.5 |
| 6. Wine Storage/Conservation in Stainless-Steel Vats | 580.0 | 502.4 | 7.2 | 580.0 | 502.4 | 7.2 |
| 7. Wine Storage/Ageing in Wooden Barrels | 702.9 | 530.3 | 7.6 | 702.9 | 530.3 | 7.6 |
| 8. Wine Heating (Madeira Wine Production) | 277.5 | 460.5 | 6.6 | — | — | — |
| 9. Batches Preparation | 582.3 | 516.4 | 7.4 | 582.3 | 516.4 | 7.4 |
| 10. Bottling | 555.0 | 411.7 | 5.9 | 555.0 | 411.7 | 5.9 |
| 11. Packaging Materials Warehouse | 293.6 | 328.0 | 4.7 | 293.6 | 328.0 | 4.7 |
| 12. Semi-Finished Product Warehouse | 241.7 | 404.7 | 5.8 | 241.7 | 404.7 | 5.8 |
| 13. Finished Product Warehouse | 346.7 | 404.7 | 5.8 | 346.7 | 404.7 | 5.8 |
| 14. Packaging | 394.0 | 362.9 | 5.2 | 394.0 | 362.9 | 5.2 |
| 15. Expedition | 481.2 | 390.8 | 5.6 | 481.2 | 390.8 | 5.6 |

The production layout was optimized based on the results obtained through the general and ergonomic risk assessments conducted for the four scenarios summarized in Table 4.

Once the optimization model was applied with the aim of reorganizing the productive areas in order to mitigate occupational risks, allying, if possible, with the operative efficiency, the results are shown in a simplified way, as uniform squares distributed along a rectangular layout, without relating directly to the various stages of production with different sizes shown in the winery layout example (Figure 3).

### 3.1. Results for Scenarios 1 and 2

According to the established scenarios 1 and 2, after applying the model to optimize the layout of the production area, several solutions were obtained, among which those that presented a minimization of general and ergonomic risks combined with the best degree of operability were selected. Thus, through Figure 4, it is possible to observe the selected solutions contemplating the execution of all operations including exclusive high-season operations (during the harvest).

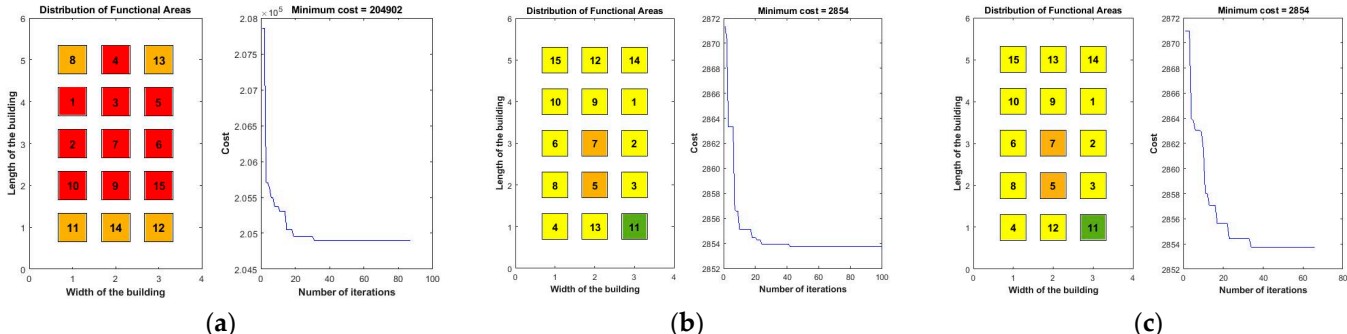

**Figure 4.** (**a**) Scenario 1; (**b**) scenario 2; (**c**) scenario 2. Solutions obtained to optimize a production area layout in terms of general and ergonomic risk reduction, including the wine heating process (production of Madeira wine).

Through Figure 4a, it is possible to observe that the zones with a higher degree of danger were redistributed along the central area, while the remaining were relocated along the periphery of the building, namely, at the southern and northern ends, with regard to the layout orientation.

Concerning efficiency, in terms of risk prevention and operational fluidity, it is possible to conclude that the simulation obtained for scenario 1 presents an efficient arrangement of zones, highlighting the proximity and contiguous location of zones 1–5, allowing for an efficient transformation of raw materials, providing a desirable distance between the transformation zones, the staging zones (zones 6–8), and the areas associated with the bottling and product finishing processes, which is required given the need to separate these steps of production for food safety reasons and for occupational safety reasons as well, favouring the degree of danger reduction.

With regard to the areas associated with bottling, packaging, and shipping, there is a dispersion of areas favourable to the efficiency of the process, once after the blending process, wines are directly transferred to the bottling area.

On the other hand, the efficiency of the subsequent steps in the bottling process is also assured, and after bottling, the product is transported directly to the packaging/palletizing area, and it is also possible to transport it directly to the expedition area, an aspect that proves to be very useful when there is production of large quantities of product for immediate expedition.

Through the analysis of Figure 4b,c, it is also verified that the zones that present a higher level of danger were redistributed along the central area, while the remaining zones were relocated along the periphery of the building.

Regarding efficiency, in terms of process operation and ergonomic risk reduction, the simulations (b) and (c) are efficient, and the choice for implementation should fall on the solution that best meets the requirements of the production process. That is, if the production process is more focused on customizing the finished product, simulation (b) will be more appropriate, since the semi-finished product warehouse (zone 12) is located near the bottling area (zone 10), the area of packaging (zone 14), and the shipping area (zone 15).

In the case of a production process of large quantities of products with a low level of variety or customization, the simulation (c) will be more efficient once the finished product warehouse (zone 13) is located in the vicinity of the area of bottling (zone 10), the packaging area (zone 14), and the dispatch area (zone 15).

In both simulations (b) and (c), the packaging materials warehouse is further away from the bottling area, which can be easily overcome by good planning and timely transfer of materials. On the other hand, the staging areas are located close to the batching area (zone 9) which, in turn, is located adjacent to the bottling area (zone 10), thus closing the production cycle.

### 3.2. Results for Scenarios 3 and 4

Similarly, through the application of the optimization model to the layout of the production area, several solutions were obtained for the scenarios 3 and 4. Thus, through Figure 5, it is possible to observe the solutions that present a minimization of general and ergonomic risks combined with the highest degree of operability, including the execution of all operations (with the wine heating process exception), including the exclusive high-season operations (during the harvest).

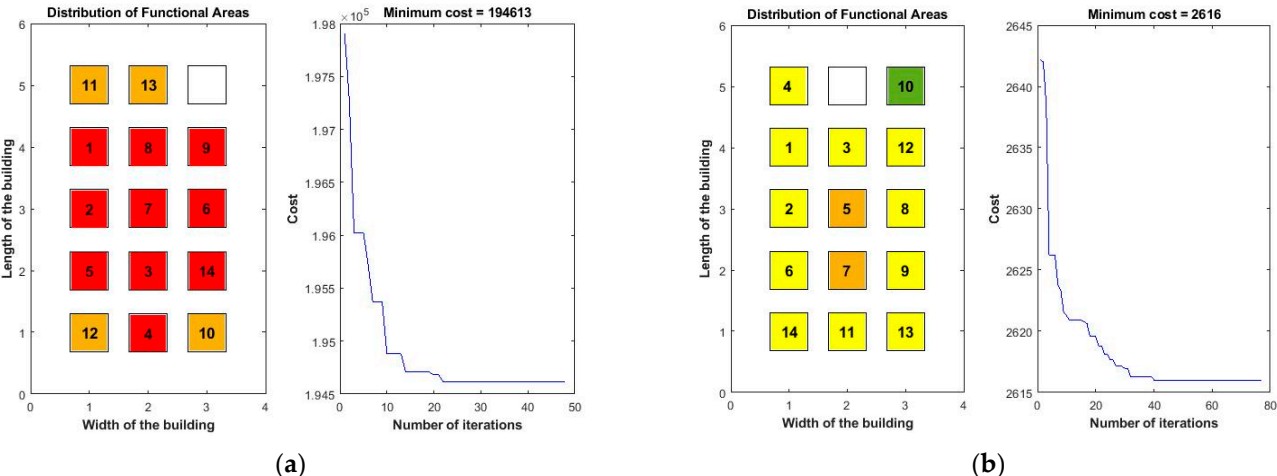

**Figure 5.** (**a**) Scenario 3; (**b**) scenario 4. Solutions obtained to optimize a production area layout in terms of general and ergonomic risk reduction, excluding the wine heating process (production of Madeira wine).

From the analysis of Figure 5a, it is possible to verify that after the application of the optimization model, the redistribution of the zones follows a similar profile to that verified in the previous scenarios, with the zones that have a higher degree of danger being redistributed along the centre of the building, while the remaining are relocated along the peripheral areas, specifically, at the south and north ends, and in relation to the orientation of the layouts.

With regard to the operational efficiency combined with the prevention of general risks, it can be seen that the selected simulation was balanced, calling attention to the proximity and contiguous location of zones 1–5, making the raw material transformation process more efficient.

In this solution, there is a desirable concentration of the ageing zones (zones 6 and 7) with the batch preparation area (zone 8) which, in turn, is located next to the bottling area (zone 9). Although the storage areas for packaging materials and finished products (zones 10 and 12) present a significant distance from the bottling area. The areas for semi-finished product storage (area 11), packaging (zone 13), and expedition (zone 14) are located in an acceptable distance from the bottling area, favouring the reduction in danger and increasing the fluidity of the production process.

Through the analysis of Figure 5b, it was also verified that the zones that presented a higher level of danger were redistributed along the central area, while the remaining were relocated along the perimeter of the building.

With regards to efficiency, in terms of process fluidity and reduction of ergonomic risk, the selected simulation shows a high level of efficiency inasmuch as the bottling area (zone 9), the blending area (zone 8), the semi-finished product storage (zone 11), the packaging (zone 13), and the expedition area (zone 14) are organized, favouring the fluidity of production activities and the reduction of potential ergonomic risks as well.

On the other hand, the areas related to the reception of raw material and the respective transformation (zones 1, 2, and 3) are close to each other, flanked by the storage/ageing

(zones 6 and 7), stabilization (zone 4) and filtering areas (zone 5), thus benefiting the production flow in the first stage.

The less positive aspect is the location of the packaging materials warehouse, which is further away from the bottling area, a fact that, however, can be easily transposed through good planning and the timely transfer of materials.

Figure 6 shows the transposition of best result obtained (Figure 4c), in terms of occupational risk mitigation and higher operative efficiency, to the example of an already existing winery (Figure 3).

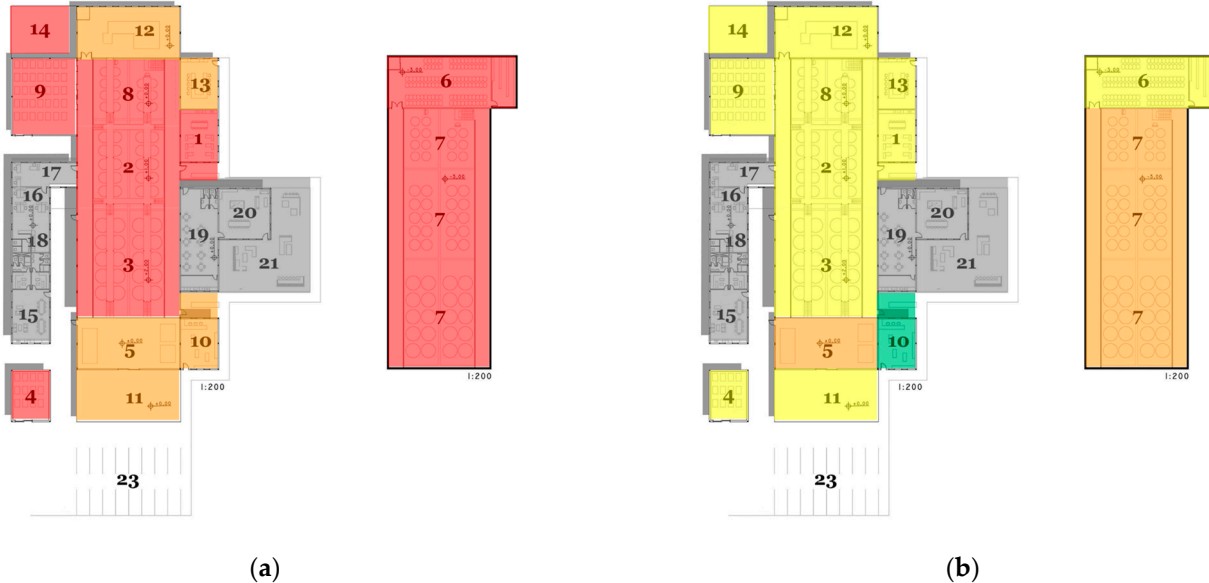

(**a**)           (**b**)

**Figure 6.** (**a**) Model based on the general risk assessment; (**b**) model based on the ergonomic risk assessment. The best model result transposition to the example of a winery layout. (1: Area of reception of raw materials (i.e., grapes); 2: winemaking area (vinification); 3: fermentation area; 4: area of clarification/stabilization; 5: filtering area; 6: wine storage/conservation in stainless−steel vats; 7: wine storage/ageing in wooden barrels; 8: batches working area; 9: bottling area; 10: warehouse for packaging materials; 11: semi−finished product warehouse; 12: finished product warehouse; 13: packaging area; 14: expedition area; 15: cafeteria/social area for employees/sanitary facilities; 16: offices; 17: technical area (oenology); 18: laboratory; 19: workshop area and shopping area (public sales); 20: restaurant; 21: terrace (outdoor); 23: car park).

Through Figure 6, it is possible to verify that the model allowed for the retrofitting of an existing winery, despite the few adaptations made due to the specific configuration of the building, namely, the zones 1–3 that were repositioned to the inner area of the building, and the zones 6 and 7, which were already located in the second floor of the main facilities. The rearrangement of production areas provided a good fit within the boundaries of the existing winery, allowing for the use of existing spaces.

## 4. Discussion

With the application of the optimization method, based on the differentiation of the two types of risk assessment: the assessment of general occupational risks and the assessment of ergonomic risks, based on the estimation of metabolic energy expenditure, it was found that the reorganization processes of areas occurred in a similar way in both cases, with the zones with a higher degree of danger tending to be relocated in the innermost areas of the building according to their respective risk levels.

With regard to the effects of the presence of the wine heating area, used for the specific production of Madeira wine, there were no significant effects in the process of

optimizing/reorganizing work areas, in both scenarios in which the general risk assessment was considered and, in the scenarios, where the ergonomic risks were considered.

It should be noted that in each scenario, the simulations performed converged to the same minimum value, naturally varying in the case of inclusion or exclusion of the wine heating process, with higher costs in the scenarios that included the wine heating process, as would be expected. It should be also noted that the application of the optimization model to both scenarios (with and without the wine heating process) allowed to obtain at least one viable solution, allying the operational efficiency and the minimization of occupational risks.

Concerning the optimization model applied to the scenarios based on the assessment of ergonomic risks with the inclusion of the wine heating process, two alternative solutions were obtained: simulation (b), which was more focused on mass production with low product variety, and simulation (c), which was more compatible with the type of production that has a greater variety of products.

In addition, the optimization model allowed for the reconfiguration of an existing winery, despite the potential adaptations that may be needed due to the specific configurations of the existing facilities, allowing this way the occupational risk mitigation providing a high operative efficiency.

The present study was essentially based on the literature review performed, both in terms of theoretical framework and in relation to practical application. For this reason, the main limitations resulting from this study are related to the use of data obtained by estimation, which were based on previous works by several authors, namely with regard to the survey and assessment of occupational risks, as well as in relation to the compilation of operations that integrate the various production processes, and these data need to be updated and confirmed in terms of practical applicability.

As future work, firstly, it is pertinent to update and confirm the data referring to the survey and assessment of occupational risks, as well as the operations that integrate the various production processes in terms of practical applicability, suggesting a practical approach in a work environment of one or several organizations to assess all production methodologies, the respective integral operations and the associated risks, both in the exercise of activities and in relation to the physical areas where the activities are carried out.

Another pertinent aspect concerns the application of other optimization models based on other algorithms, which could provide solutions based on scenarios with zones or departments of different sizes, establishing a comparison between the results obtained, namely if there is any potential effect related to the size of zones that could affect the facilities reorganization and the respective levels of danger.

**Author Contributions:** Conceptualization, A.A.F. and T.M.L.; methodology, T.M.L. and P.D.G.; software, A.A.F.; validation T.M.L. and P.D.G.; formal analysis, T.M.L.; investigation, A.A.F.; resources, T.M.L. and P.D.G.; data curation, A.A.F.; writing—original draft preparation, A.A.F.; writing—review and editing, T.M.L. and P.D.G.; supervision, T.M.L. All authors have read and agreed to the published version of the manuscript.

**Funding:** This research received no external funding.

**Institutional Review Board Statement:** Not applicable.

**Informed Consent Statement:** Not applicable.

**Data Availability Statement:** The authors confirm that the data supporting the findings of this study are available within the article.

**Acknowledgments:** This work was supported in part by the Fundação para a Ciência e Tecnologia (FCT) and C-MAST (Centre for Mechanical and Aerospace Science and Technologies), under project UIDB/00151/2020.

**Conflicts of Interest:** The authors declare no conflict of interest.

## Appendix A

**Table A1.** Estimates of the safety criteria, assessment, and quantification of the general risks.

| Risks | Safety Criteria | | | Assessment | |
|---|---|---|---|---|---|
| | Exposure Factor | Consequence Factor | Probability Factor | Hazard Degree | Risk Index |
| **Reception of Raw Materials (i.e., Grapes)** | | | | | |
| Musculoskeletal injuries (back lumbar) | 10 | 5 | 10 | 500 | Very High |
| Visual fatigue | 10 | 5 | 6 | 300 | High |
| Physical fatigue | 10 | 5 | 10 | 500 | Very High |
| Falling objects/materials | 10 | 5 | 6 | 300 | High |
| Falls from height | 3 | 15 | 6 | 270 | High |
| Falls at the same level | 6 | 5 | 10 | 300 | High |
| Shocks against objects | 6 | 5 | 10 | 300 | High |
| Jams | 6 | 15 | 6 | 540 | Very High |
| Crushes | 6 | 25 | 6 | 900 | Very High |
| Run over | 6 | 25 | 6 | 900 | Very High |
| Forklift rollover | 10 | 25 | 3 | 750 | Very High |
| Noise exposure | 10 | 5 | 10 | 500 | Very High |
| Exposure to vibrations | 10 | 5 | 6 | 300 | High |
| Electrical hazards | 2 | 15 | 6 | 180 | Remarkable |
| **Average hazard degree** | | | | **467.1** | **Very High** |
| **Vinification** | | | | | |
| Physical fatigue | 10 | 5 | 10 | 500 | Very High |
| Falls at the same level | 6 | 5 | 6 | 180 | Remarkable |
| Falls from height | 2 | 15 | 6 | 180 | Remarkable |
| Shocks against objects | 6 | 5 | 10 | 300 | High |
| Exposure to toxic gases ($CO_2$, $SO_2$) | 10 | 15 | 10 | 1500 | Very High |
| Exposure to carcinogens | 6 | 15 | 10 | 900 | Very High |
| Exposure to harmful chemicals | 6 | 15 | 6 | 540 | Very High |
| Electrical hazards | 6 | 15 | 3 | 270 | Remarkable |
| Breathing difficulties | 6 | 15 | 6 | 540 | Very High |
| Exposure to hypoxic environments (caused by low concentrations of atmospheric oxygen) | 6 | 25 | 10 | 1500 | Very High |
| Jams | 3 | 15 | 3 | 135 | Remarkable |
| Musculoskeletal injuries | 6 | 5 | 10 | 300 | High |
| Noise exposure | 10 | 15 | 6 | 900 | Very High |
| Exposure to vibrations | 10 | 5 | 6 | 300 | High |
| **Average hazard degree** | | | | **574.6** | **Very High** |

**Table A1.** *Cont.*

| Risks | Safety Criteria | | | Assessment | |
|---|---|---|---|---|---|
| | Exposure Factor | Consequence Factor | Probability Factor | Hazard Degree | Risk Index |
| **Fermentation** | | | | | |
| Physical fatigue | 10 | 5 | 10 | 500 | Very High |
| Falls at the same level | 6 | 5 | 6 | 180 | Remarkable |
| Falls from height | 6 | 15 | 6 | 540 | Very High |
| Jams | 3 | 15 | 3 | 135 | Remarkable |
| Shocks against objects | 6 | 5 | 10 | 300 | High |
| Exposure to toxic gases ($CO_2$, $SO_2$) | 10 | 15 | 10 | 1500 | Very High |
| Exposure to carcinogens | 6 | 15 | 10 | 900 | Very High |
| Exposure to harmful chemicals | 6 | 15 | 6 | 540 | Very High |
| Exposure to hypoxic environments (caused by low concentrations of atmospheric oxygen) | 6 | 25 | 10 | 1500 | Very High |
| Musculoskeletal injuries | 6 | 5 | 10 | 300 | High |
| Formation of explosive atmospheres | 3 | 25 | 3 | 225 | High |
| Breathing difficulties | 6 | 15 | 6 | 540 | Very High |
| Electrical hazards | 6 | 15 | 3 | 270 | High |
| **Average hazard degree** | | | | **571.5** | **Very High** |
| **Clarification/Stabilization** | | | | | |
| Physical fatigue | 10 | 5 | 6 | 300 | High |
| Falls at the same level | 6 | 5 | 6 | 180 | Remarkable |
| Falls from height | 6 | 15 | 6 | 540 | Very High |
| Shocks against objects | 6 | 5 | 6 | 180 | Remarkable |
| Exposure to carcinogens | 6 | 15 | 6 | 540 | Very High |
| Exposure to harmful chemicals | 6 | 15 | 6 | 540 | Very High |
| Electrical hazards | 6 | 15 | 6 | 540 | Very High |
| Breathing difficulties | 10 | 15 | 6 | 900 | Very High |
| Musculoskeletal injuries | 6 | 5 | 6 | 180 | Remarkable |
| Jams | 3 | 15 | 3 | 135 | Remarkable |
| Noise exposure | 10 | 15 | 6 | 900 | Very High |
| Exposure to vibrations | 10 | 5 | 6 | 300 | High |
| **Average hazard degree** | | | | **462.9** | **Very High** |
| **Wine Storage/Conservation in Stainless-Steel Vats** | | | | | |
| Falls from height | 10 | 15 | 6 | 900 | Very High |
| Falls at the same level | 10 | 5 | 6 | 300 | High |
| Musculoskeletal injuries | 10 | 5 | 10 | 500 | Very High |
| Jams | 6 | 15 | 3 | 270 | High |
| Shocks against objects | 6 | 5 | 6 | 180 | Remarkable |
| Exposure to vapours | 6 | 15 | 6 | 540 | Very High |
| Exposure to hazardous substances | 6 | 15 | 6 | 540 | Very High |

**Table A1.** *Cont.*

| Risks | Safety Criteria | | | Assessment | |
|---|---|---|---|---|---|
| | Exposure Factor | Consequence Factor | Probability Factor | Hazard Degree | Risk Index |
| Formation of explosive atmospheres | 6 | 50 | 6 | 1800 | Very High |
| Breathing difficulties | 6 | 15 | 6 | 540 | Very High |
| Dental erosion | 6 | 15 | 6 | 540 | Very High |
| Electrical hazards | 6 | 15 | 3 | 270 | High |
| **Average hazard degree** | | | | **580** | **Very High** |
| **Wine Storage/Ageing in Wooden Barrels** | | | | | |
| Falls from height | 10 | 15 | 6 | 900 | Very High |
| Falls at the same level | 10 | 5 | 6 | 300 | High |
| Musculoskeletal injuries (back lumbar) | 10 | 15 | 10 | 1500 | Very High |
| Jams | 3 | 15 | 6 | 270 | High |
| Shocks against objects | 6 | 5 | 6 | 180 | Remarkable |
| Electrical hazards | 3 | 15 | 6 | 270 | High |
| Crushes | 10 | 25 | 6 | 1500 | Very High |
| **Average Hazard Degree** | | | | **702.9** | **Very High** |
| **Wine Heating (Madeira Wine Production)** | | | | | |
| Falls from height | 3 | 15 | 6 | 270 | High |
| Falls at the same level | 3 | 5 | 6 | 90 | Remarkable |
| Musculoskeletal injuries (back lumbar) | 3 | 15 | 6 | 270 | High |
| Jams | 3 | 15 | 6 | 270 | High |
| Shocks against objects | 3 | 5 | 6 | 90 | Remarkable |
| Exposure to vapours | 3 | 15 | 6 | 270 | High |
| Exposure to hazardous substances | 3 | 15 | 6 | 270 | High |
| Formation of explosive atmospheres | 3 | 50 | 6 | 900 | Very High |
| Breathing difficulties | 3 | 15 | 6 | 270 | High |
| Thermal stress | 3 | 5 | 6 | 90 | Remarkable |
| Burns | 3 | 15 | 6 | 270 | High |
| Electrical hazards | 3 | 15 | 6 | 270 | High |
| **Average Hazard Degree** | | | | **277.5** | **High** |
| **Batches Preparation** | | | | | |
| Falls from height | 10 | 15 | 6 | 900 | Very High |
| Falls at the same level | 10 | 5 | 6 | 300 | High |
| Musculoskeletal injuries (back lumbar) | 10 | 15 | 6 | 900 | Very High |
| Jams | 3 | 15 | 3 | 135 | Remarkable |
| Shocks against objects | 10 | 5 | 6 | 300 | High |
| Exposure to vapours | 3 | 15 | 6 | 270 | High |
| Exposure to hazardous substances | 6 | 15 | 6 | 540 | Very High |
| Formation of explosive atmospheres | 6 | 50 | 6 | 1800 | Very High |

**Table A1.** *Cont.*

| Risks | Safety Criteria | | | Assessment | |
|---|---|---|---|---|---|
| | Exposure Factor | Consequence Factor | Probability Factor | Hazard Degree | Risk Index |
| Breathing difficulties | 6 | 15 | 6 | 540 | Very High |
| Dental erosion | 6 | 5 | 6 | 180 | Remarkable |
| Electrical hazards | 6 | 15 | 6 | 540 | Very High |
| **Average Hazard Degree** | | | | 582.3 | **Very High** |
| **Bottling** | | | | | |
| Jams | 10 | 15 | 6 | 900 | Very High |
| Crushes | 3 | 25 | 6 | 450 | Very High |
| Cuts | 10 | 15 | 6 | 900 | Very High |
| Exposure to vapours | 3 | 15 | 6 | 270 | High |
| Visual fatigue | 10 | 5 | 10 | 500 | Very High |
| Physical fatigue | 10 | 5 | 10 | 500 | Very High |
| Noise exposure | 10 | 15 | 10 | 1500 | Very High |
| Exposure to vibrations | 10 | 5 | 10 | 500 | Very High |
| Falls from height | 3 | 15 | 6 | 270 | High |
| Falls at the same level | 10 | 5 | 6 | 300 | High |
| Load drop | 10 | 15 | 6 | 900 | Very High |
| Musculoskeletal injuries | 6 | 15 | 6 | 540 | Very High |
| Biological hazards | 3 | 5 | 3 | 45 | Moderate |
| Non-ionizing radiation (exposure to ultraviolet radiation; inflammation of the tissues of the eyeball and skin burns) | 6 | 15 | 6 | 540 | Very High |
| Non-ionizing radiation (infrared radiation: skin burns, persistent increase in skin pigmentation, and eye damage) | 6 | 15 | 6 | 540 | Very High |
| Non-ionizing radiation (laser: ocular corneal burn, severe retinal injury, or skin burns) | 6 | 15 | 6 | 540 | Very High |
| Electrical hazards | 6 | 15 | 6 | 540 | Very High |
| Exposure to vapours | 3 | 15 | 6 | 270 | High |
| **Average hazard degree** | | | | 555.8 | **Very High** |
| **Packaging Materials Warehouse** | | | | | |
| Musculoskeletal injuries | 6 | 15 | 6 | 540 | Very High |
| Visual fatigue | 6 | 5 | 10 | 300 | High |
| Load drop | 6 | 15 | 6 | 540 | Very High |
| Electrical hazards | 3 | 15 | 3 | 135 | Remarkable |
| Run over | 2 | 25 | 6 | 300 | High |
| Forklift rollover | 2 | 25 | 3 | 150 | Remarkable |
| Exposure to vibrations | 3 | 5 | 6 | 90 | Remarkable |
| **Average hazard degree** | | | | 293.6 | **High** |

**Table A1.** *Cont.*

| Risks | Safety Criteria | | | Assessment | |
|---|---|---|---|---|---|
| | Exposure Factor | Consequence Factor | Probability Factor | Hazard Degree | Risk Index |
| **Semi-Finished Product Warehouse** | | | | | |
| Musculoskeletal injuries | 6 | 5 | 6 | 180 | Remarkable |
| Run over | 6 | 25 | 3 | 450 | Very High |
| Visual fatigue | 6 | 5 | 6 | 180 | Remarkable |
| Physical fatigue | 6 | 5 | 6 | 180 | Remarkable |
| Load drop | 3 | 15 | 6 | 270 | High |
| Forklift rollover | 2 | 25 | 3 | 150 | Remarkable |
| Exposure to vibrations | 3 | 5 | 6 | 90 | Remarkable |
| Cuts | 6 | 15 | 6 | 540 | Very High |
| Electrical hazards | 3 | 15 | 3 | 135 | Remarkable |
| **Average hazard degree** | | | | **241.7** | **High** |
| **Finished Product Warehouse** | | | | | |
| Musculoskeletal injuries | 10 | 5 | 6 | 300 | High |
| Run over | 6 | 25 | 3 | 450 | Very High |
| Visual fatigue | 10 | 5 | 6 | 300 | High |
| Physical fatigue | 6 | 5 | 6 | 180 | Remarkable |
| Load drop | 6 | 15 | 6 | 540 | Very High |
| Forklift rollover | 6 | 25 | 3 | 450 | Very High |
| Exposure to vibrations | 6 | 5 | 3 | 90 | Remarkable |
| Cuts | 6 | 15 | 6 | 540 | Very High |
| Electrical hazards | 3 | 15 | 6 | 270 | High |
| **Average hazard degree** | | | | **346.7** | **High** |
| **Packaging** | | | | | |
| Cuts | 6 | 15 | 6 | 540 | Very High |
| Jams | 6 | 15 | 6 | 540 | Very High |
| Crushes | 3 | 25 | 6 | 450 | Very High |
| Falls from height | 6 | 15 | 6 | 540 | Very High |
| Falls at the same level | 10 | 5 | 6 | 300 | High |
| Musculoskeletal injuries (back lumbar) | 10 | 5 | 6 | 300 | High |
| Load drop | 6 | 15 | 6 | 540 | Very High |
| Run over | 3 | 25 | 3 | 225 | High |
| Load drop | 6 | 15 | 6 | 540 | Very High |
| Forklift rollover | 3 | 25 | 3 | 225 | High |
| Exposure to vibrations | 1 | 5 | 6 | 30 | Moderate |
| Noise exposure | 6 | 15 | 6 | 540 | Very High |
| Visual fatigue | 10 | 5 | 6 | 300 | High |
| Physical fatigue | 10 | 5 | 6 | 300 | High |
| Electrical hazards | 6 | 15 | 6 | 540 | Very High |
| **Average hazard degree** | | | | **394** | **High** |

**Table A1.** *Cont.*

| Risks | Safety Criteria | | | Assessment | |
|---|---|---|---|---|---|
| | Exposure Factor | Consequence Factor | Probability Factor | Hazard Degree | Risk Index |
| Expedition | | | | | |
| Musculoskeletal injuries (back lumbar) | 6 | 5 | 6 | 180 | Remarkable |
| Physical fatigue | 6 | 5 | 6 | 180 | Remarkable |
| Visual fatigue | 6 | 15 | 6 | 540 | Very High |
| Falls from height | 6 | 15 | 6 | 540 | Very High |
| Falls at the same level | 6 | 5 | 6 | 180 | Remarkable |
| Run over | 6 | 25 | 6 | 900 | Very High |
| Load drop | 6 | 15 | 6 | 540 | Very High |
| Forklift rollover | 6 | 25 | 6 | 900 | Very High |
| Crushes | 6 | 25 | 6 | 900 | Very High |
| Jams | 6 | 15 | 6 | 540 | Very High |
| Noise exposure | 3 | 15 | 6 | 270 | High |
| Exposure to vibrations | 3 | 5 | 3 | 45 | Moderate |
| Cuts | 6 | 15 | 6 | 540 | Very High |
| **Average hazard degree** | | | | **481.2** | **Very High** |

## Appendix B

**Table A2.** Summary of the activities involved in the production processes of the wine sector, with reference to the main tasks involved and the respective estimates of expenditure of metabolic energy.

| Activities | Main Tasks | Estimation of Metabolic Energy Expenditure | | | Level |
|---|---|---|---|---|---|
| | | MET | J/s | kcal/min | |
| Grape Reception and Unloading | Vehicle unloading/dumping of boxes with grapes (Manual handling of loads) | 6.0 | 523.4 | 7.5 | Heavy |
| | Heavy work in a standing position | 4.5 | 390.8 | 5.6 | Moderate |
| | Forklift driving | 2.5 | 216.3 | 3.1 | Light |
| | Going up and down from the cargo box of vehicles/tractor with trailer (work on uneven ground) | 8.8 | 767.6 | 11.0 | Very Heavy |
| | Going down and upstairs to access the hopper for maintenance/cleaning of the hopper (unlevelled work) | 7.5 | 655.9 | 9.4 | Heavy |
| | **Average:** | **5.9** | 509.4 | **7.3** | **Moderate** |
| Grape Selection/Sorting | Handling of boxes with rejected grapes (manual handling of loads) | 6.5 | 565.2 | 8.1 | Very Heavy |
| | Moderate work in a standing position | 3.5 | 307.0 | 4.4 | Light |
| | **Average:** | **5.0** | **439.6** | **6.3** | **Moderate** |
| Destemming/Crushing | Opening the destemming/crushing and disassembly system/assembly of components for clearing/maintenance/cleaning (manual handling of loads) | 4.5 | 390.8 | 5.6 | Moderate |

| Activities | Main Tasks | Estimation of Metabolic Energy Expenditure | | | Level |
|---|---|---|---|---|---|
| | | MET | J/s | kcal/min | |
| **Destemming/Crushing** | Removal of the stalk (manual handling of loads) | 8.0 | 697.8 | 10.0 | Very Heavy |
| | Going up and down stairs to access the extraction system/ducts for maintenance/cleaning of the extraction system/ducts (work on uneven levels) | 5.8 | 509.4 | 7.3 | Moderate |
| | Maintenance/cleaning of the extraction system/ducts | 4.5 | 390.8 | 5.6 | Moderate |
| | Moderate work in a standing position | 3.5 | 307.0 | 4.4 | Light |
| | **Average:** | **5.3** | **460.5** | **6.6** | **Moderate** |
| **Sulphiting** | Superior access to the vats (unlevelled work—going up and down stairs) | 5.8 | 509.4 | 7.3 | Moderate |
| | Moderate work in a standing position | 3.5 | 307.0 | 4.4 | Light |
| | **Average:** | **4.7** | **411.7** | **5,9** | **Moderate** |
| **Must Clarification** | Transport of filter materials (manual handling of loads) | 8.0 | 697.8 | 10.0 | Very Heavy |
| | Removal of filter residues (manual handling of loads) | 8.0 | 697.8 | 10.0 | Very Heavy |
| | Transfer operations: handling of transfer pumps, hoses and connections (manual handling of loads) | 8.0 | 697.8 | 10.0 | Very Heavy |
| | Superior access to the vats (unlevelled work—going up and down stairs) | 5.8 | 509.4 | 7.3 | Moderate |
| | Heavy work in a standing position | 4.5 | 390.8 | 5.6 | Moderate |
| | **Average:** | **6.9** | **600.1** | **8,6** | **Moderate** |
| **Must Preparation** | Cleaning/preparation of fermentation vats (manual handling of loads) | 8.0 | 697.8 | 10.0 | Very Heavy |
| | Superior access to the vats (unlevelled work—going up and down stairs) | 5.8 | 509.4 | 7.3 | Moderate |
| | Moderate work in a standing position | 3.5 | 307.0 | 4.4 | Light |
| | **Average:** | **5.8** | **502.4** | **7,2** | **Moderate** |
| **Control of Alcoholic Fermentation** | Superior access to the vats (unlevelled work—going up and down stairs) | 5.8 | 509.4 | 7.3 | Moderate |
| | Fermentation control | 4.2 | 369.8 | 5.3 | Light |
| | **Average:** | **5.0** | **439.6** | **6,3** | **Moderate** |
| **Maceration Verification/Monitoring** | Superior access to the vats (unlevelled work—going up and down stairs) | 5.8 | 509.4 | 7.3 | Moderate |
| | Moderate work in a standing position | 3.5 | 307,0 | 4.4 | Very Heavy |
| | **Average:** | **4.7** | **411.7** | **5,9** | **Moderate** |
| **Malolactic Fermentation Control (White Wines)** | Transfer operations: handling of transfer pumps, hoses and connections (manual handling of loads) | 8.0 | 697.8 | 10.0 | Very Heavy |
| | Superior access to the vats (unlevelled work—going up and down stairs) | 5.8 | 509.4 | 7.3 | Moderate |
| | Moderate work in a standing position | 3.5 | 307.0 | 4.4 | Light |

**Table A2.** *Cont.*

| Activities | Main Tasks | Estimation of Metabolic Energy Expenditure | | | Level |
|---|---|---|---|---|---|
| | | **MET** | **J/s** | **kcal/min** | |
| | **Average:** | **5.8** | **502.4** | **7.2** | **Moderate** |
| **Stabilization/ Clarification** | Operate/control stabilization/clarification system | 2.5 | 216.3 | 3.1 | Light |
| | Material handling for stabilization/clarification (manual handling of loads) | 8.0 | 697.8 | 10.0 | Very Heavy |
| | Forklift driving | 2.5 | 216.3 | 3.1 | Light |
| | Superior access to the vats (unlevelled work—going up and down stairs) | 5.8 | 509.4 | 7.3 | Moderate |
| | Moderate work in a standing position | 3.5 | 307.0 | 4.4 | Light |
| | **Average:** | **4.5** | **390.8** | **5.6** | **Moderate** |
| **Filtering** | Transport of materials for filtering (manual handling of loads) | 8.0 | 697.8 | 10.0 | Very Heavy |
| | Removal of filter residues (manual handling of loads) | 8.0 | 697.8 | 10.0 | Very Heavy |
| | Forklift driving | 2.5 | 216.3 | 3.1 | Light |
| | Transfer operations: handling of transfer pumps, hoses, and connections (manual handling of loads) | 8.0 | 697.8 | 10.0 | Heavy |
| | Superior access to the vats (unlevelled work—going up and down stairs) | 5.8 | 509.4 | 7.3 | Moderate |
| | Moderate work in a standing position | 3.5 | 307.0 | 4.4 | Light |
| | **Average:** | **6.0** | **523.4** | **7.5** | **Moderate** |
| **Sulphiting** | Superior access to the vats (unlevelled work—going up and down stairs) | 5.8 | 509.4 | 7.3 | Moderate |
| | Moderate work in a standing position | 3.5 | 307.0 | 4.4 | Light |
| | **Average:** | 4.7 | 411.7 | 5.9 | Moderate |
| **Storage/Ageing (Wooden Barrels)** | Preparation of wooden barrels | **7** | **614.1** | **8.8** | **Moderate** |
| | Forklift driving (transport of wooden barrels) | 2.5 | 216.3 | 3.1 | Light |
| | Packaging/stacking of wooden barrels in construction sites/metal supports (manual handling of loads) | 7.5 | 655.9 | 9.4 | Moderate |
| | Transfer operations: handling of transfer pumps, hoses, and connections (manual handling of loads) | 8 | 697.8 | 10 | Very Heavy |
| | Superior access to the vats (unlevelled work—going up and down stairs) | 5.8 | 509.4 | 7.3 | Moderate |
| | Access to rows of barrels on the higher levels (unlevelled work—going up and down stairs) | 5.8 | 509.4 | 7.3 | Moderate |
| | Heavy work in a standing position | 4.5 | 390.8 | 5.6 | Moderate |
| | Unstacking of wooden barrels in construction site/metal supports after the ageing process (manual handling of loads) | 7.5 | 655.9 | 9.4 | Heavy |

**Table A2.** *Cont.*

| Activities | Main Tasks | Estimation of Metabolic Energy Expenditure | | | Level |
|---|---|---|---|---|---|
| | | MET | J/s | kcal/min | |
| | Average: | 6.1 | 530.3 | 7.6 | Moderate |
| **Wine Storage/Conservation in Stainless-Steel Vats** | Cleaning/preparation of stainless-steel vats (Manual handling of loads) | 8 | 697.8 | 10 | Very Heavy |
| | Transfer operations: handling of transfer pumps, hoses, and connections (manual handling of loads) | 8 | 697.8 | 10 | Very Heavy |
| | Superior access to the vats (unlevelled work—going up and down stairs) | 5.8 | 509.4 | 7.3 | Moderate |
| | Moderate work in a standing position | 3.5 | 307.0 | 4.4 | Light |
| | Average: | 6.3 | 551.3 | 7.9 | Moderate |
| **Wine Heating (Madeira Wine Production)** | Cleaning/preparation of stainless-steel vats (manual handling of loads) | 8 | 697.8 | 10 | Very Heavy |
| | Transfer operations: handling of transfer pumps, hoses and connections (Manual handling of loads) | 8 | 697.8 | 10 | Very Heavy |
| | Superior access to the vats (unlevelled work—going up and down stairs) | 5.8 | 509.4 | 7.3 | Moderate |
| | Maintenance/checking of piping for water circulation for heating (manual handling of loads) | 3 | 265.2 | 3.8 | Light |
| | Maintenance/checking of heating boilers (manual handling of loads) | 3 | 265.2 | 3.8 | Light |
| | Moderate work in a standing position | 3.5 | 307.0 | 4.4 | Light |
| | Average: | 5.2 | 460.5 | 6.6 | Moderate |
| **Batches Preparation (Blending)** | Cleaning/preparation of stainless-steel vats (manual handling of loads) | 8 | 697.8 | 10 | Very Heavy |
| | Transfer operations: handling of transfer pumps, hoses, and connections (manual handling of loads) | 8 | 697.8 | 10 | Very Heavy |
| | Superior access to the vats (unlevelled work—going up and down stairs) | 5.8 | 509.4 | 7.3 | Moderate |
| | Moderate work in a standing position | 3.5 | 307.0 | 4.4 | Light |
| | Average: | 6.3 | 551.3 | 7.9 | Moderate |
| **Stabilization/ Clarification** | Operate/control stabilization/clarification system | 2.5 | 216.3 | 3.1 | Light |
| | Transport of materials for stabilization/clarification (manual handling of loads) | 8 | 697.8 | 10 | Very Heavy |
| | Forklift driving | 2.5 | 216.3 | 3.1 | Light |
| | Superior access to the vats (unlevelled work—going up and down stairs) | 5.8 | 509.4 | 7.3 | Moderate |
| | Moderate work in a standing position | 3.5 | 307.0 | 4.4 | Light |
| | Average: | 4.5 | 390.8 | 5.6 | Moderate |

| Activities | Main Tasks | Estimation of Metabolic Energy Expenditure | | | Level |
|---|---|---|---|---|---|
| | | **MET** | **J/s** | **kcal/min** | |
| Filtering | Transport of materials for filtering (manual handling of loads) | 8 | 697.8 | 10 | Very Heavy |
| | Removal of filter residues (manual handling of loads) | 8 | 697.8 | 10 | Very Heavy |
| | Forklift driving | 2.5 | 216.3 | 3.1 | Light |
| | Transfer operations: handling of transfer pumps, hoses, and connections (manual handling of loads) | 8 | 697.8 | 10 | Very Heavy |
| | Superior access to the vats (unlevelled work—going up and down stairs) | 5.8 | 509.4 | 7.3 | Moderate |
| | Moderate work in a standing position | 3.5 | 307.0 | 4.4 | Leve |
| | **Average:** | **6** | **523.4** | **7.5** | **Moderate** |
| Final Corrections | Transport of oenological adjuvants/additives (manual handling of loads) | 8 | 697.8 | 10 | Very Heavy |
| | Transfer operations: handling of transfer pumps, hoses, and connections (manual handling of loads) | 8 | 697.8 | 10 | Very Heavy |
| | Superior access to the vats (unlevelled work—going up and down stairs) | 5.8 | 509.4 | 7.3 | Moderate |
| | Forklift driving | 2.5 | 216.3 | 3.1 | Light |
| | Moderate work in a standing position | 3.5 | 307.0 | 4.4 | Light |
| | **Average:** | **5.6** | **488.5** | **7.0** | **Moderate** |
| Receipt of Packaging Materials | Verification Process (unlevelled work—going up and down stairs) | 5.8 | 509.4 | 7.3 | Moderate |
| | Forklift driving | 2.5 | 216.3 | 3.1 | Light |
| | Plastic cutting and removal | 3 | 265.2 | 3.8 | Light |
| | Moderate work in a standing position | 3.5 | 307.0 | 4.4 | Light |
| | **Average:** | **3.7** | **328.0** | **4.7** | **Moderate** |
| Depalletization | Plastic cutting and removal | 3 | 265.2 | 3.8 | Light |
| | Supply of bottles to the bottling line (manual handling of loads) | 4.5 | 390.8 | 5.6 | Moderate |
| | Forklift driving | 2.5 | 216.3 | 3.1 | Light |
| | Moderate work in a standing position | 3.5 | 307.0 | 4.4 | Light |
| | **Average:** | **3.4** | **293.1** | **4.2** | **Moderate** |
| Bottles Rinsing | Operate/control bottle rinsing system | 2.5 | 216.3 | 3.1 | Light |
| | Cleaning/preparation of the bottle rinsing system (manual handling of loads) | 8 | 697.8 | 10 | Very Heavy |
| | Maintenance of the bottle rinsing system (manual handling of loads) | 3 | 265.2 | 3.8 | Light |
| | Access to the bottle rinsing system from the top (unlevelled work—going up and down stairs) | 7.5 | 655.9 | 9.4 | Heavy |
| | Moderate work in a standing position | 3.5 | 307.0 | 4.4 | Light |

**Table A2.** *Cont.*

| Activities | Main Tasks | Estimation of Metabolic Energy Expenditure | | | Level |
| --- | --- | --- | --- | --- | --- |
| | | **MET** | **J/s** | **kcal/min** | |
| | Average: | **4.9** | **425.7** | **6.1** | **Moderate** |
| **Palletizing** | Operate/control palletizing system | 2.5 | 216.3 | 3.1 | Light |
| | Pallet positioning (manual handling of loads) | 4.5 | 390.8 | 5.6 | Moderate |
| | Packaging of finished product boxes on pallets (manual handling of loads) | 8 | 697.8 | 10 | Very Heavy |
| | Forklift driving | 2.5 | 216.3 | 3.1 | Light |
| | Moderate work in a standing position | 3.5 | 307.0 | 4.4 | Light |
| | Average: | **4.2** | **362.9** | **5.2** | **Moderate** |
| **Storage** | Storage of finished product boxes (manual handling of loads) | 8 | 697.8 | 10 | Very Heavy |
| | Forklift driving | 2.5 | 216.3 | 3.1 | Light |
| | Moderate work in a standing position | 3.5 | 307.0 | 4.4 | Light |
| | Average: | **4.7** | **404.7** | **5.8** | **Moderate** |
| **Expedition** | Pallet check (unlevelled work—going up and down stairs) | 7.5 | 655.9 | 9.4 | Heavy |
| | Forklift driving | 2.5 | 216.3 | 3.1 | Light |
| | Moderate work in a standing position | 3.5 | 307.0 | 4.4 | Light |
| | Average: | **4.5** | **390.8** | **5.6** | **Moderate** |

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
