# Peer review of "Meta-Heuristic Model for Optimization of Production Layouts Based on Occupational Risk Assessment: Application to the Portuguese Wine Sector"

_asi, doi:10.3390/asi5020040_

Round 1

Reviewer 1 Report

The authors present a study on the optimization of production layouts considering occupational risks. The proposed approach is applied to the optimization of the layout of a winery. The paper is interesting, but needs some rewriting and additional explanations before it can be accepted for publication. Here are my suggestions that would hopefully improve the work and make it suitable for publication.

Minor stylistic edits:

  • Line 82: ..., [11] determined…   I suggest rewriting similar sentences to either:
  1. …, Author1 and Author2 [11] determined / …, Author 1 et al. [11] determined
  2. …, two possible layout solutions were determined [11]: …

  • Please use Word’s equation type-setting functionality or some other software to write the equations. Use symbols for variables and parameters: e.g. l instead of the word length. Indices should be subscribed (e,g. li,j) and explained.

Major comments:

  • Section 1.1. The genetic algorithm: Genetic algorithms and their inner workings are fairly well known by now. Therefore, unless there changes in the algorithmic procedures (e.g., user-defined/ new routines) that need to be known in order to repeat the procedure, I propose to shorten this section considerably or remove it entirely from the manuscript. This is critical to the manuscript because it allows the reader to focus on the actual novelty rather than spending time reading the inner workings of off-the-shelf solutions. My suggestion would be to incorporate the vital information, if any, in chapter 2 (The optimization model).
  • Chapter 2 -The optimization model: This chapter needs a considerable reworking. The best way to approach would be to see it with the eyes of the reader who wants to repeat or implement your procedure in the same environment as you did, or better yet in some other environment. My point is that this chapter lacks relationships between the data in appendix and variables/parameters in the model. My suggestion would be to structure the model in the following way:

Start with the objective/fitness function with detailed description of each variable, followed by all equality and inequality constraints that describe the feasible region of the problem.

  • Each variable/parameter in the constraints must be described as well.
  • Please provide the info on whether a given variable/parameter is integer or continuous.
  • Also, the function (the role) of the constraint should be provided (e.g., Equation XY provides the upper limit on…, Equation XZ determines the…)

In essence, the model should be written in the form:

This would make the manuscript more general, easier to reproduce in other optimization environments, and less of a case study. Note that the results of the optimization are commonly more affected by the modelling approach (definition of the objective function and feasible region) than by the solution procedure.

  • Line 223: I believe that the model is not based on the genetic algorithm, the solution procedure is, however.
  • Line 229: What relation (equation) was used to calculate these costs.
  • Line 232: The same comment as for line 229.
  • Line 244: Please use symbols for variables and parameters when writing equations (e.g., NZ=lxly, where NZ is the number of zones, lx(m) is the length of, ….).
  • Figure 6 is unnecessary, as everything can be, and partly is, explained in text. The problematic part is that is strictly oriented towards Matlab environment; thus, it gives no directly useful information to a reader working in Julia, C++, Python etc.
  • Line 282: Variables and indices should be written in italic, according to the SI standard. Also, how are CA, CG, and L calculated?  (See comment 4). What are the units of measure of CACG, and L?
  • Line 343: What role does the Euclidian distance play in the model. Is it an equality constraint, an inequality constraint, is it directly or indirectly affecting the objective function? (See comment 4)
  • Line 413: I believe the title should be “Optimization results”.
  • Line 421: It is not clear if the data in Table 4 is the result of the optimization or derived in some other way. If latter, it should be explained how it was derived. Table 4: Please use SI units. kcal is not an SI unit.
  • Lines 423–526: Results for scenarios. Comparing the layouts in Figs. 8 & 9 with that in Fig. 7, it is noticeable that the areas required for the various stages of production are different. While they are clearly different in Fig. 7, this is not the case in the optimized results (Figs. 8 & 9), where they are shown as uniform squares.

The point that interests me is the following:

  1. Is the representation in Figs. 8 & 9 the way it is just because of the simplicity of the representation? If so, how have the constraints of the different production areas been accounted for in the model? (see comment 4)
  2. Furthermore, is the model suitable for grass-roots design of the winery or for retrofitting the existing winery. If the latter, the rearranged production areas should fit within the boundaries of the existing winery as much as possible.
  3. A figure showing the actual optimal layout would be a nice addition, as it could show which assumptions should be banned in future work, as well as some new ideas for future work.

Author Response

Thank you for all your comments and suggestions.

Minor stylistic edits:

Line 82 (now Line 86): Done as suggested

Word’s equation was used, symbols were used for variables and parameters and explained.

Major comments:

Section 1.1. The genetic algorithm: The section was removed and some useful information related to the GA Implementation Flow Chart was transferred to the next chapter, which was reorganized and renamed as Material and Methods.

Chapter 2 -The optimization model (now called Material and Methods): The relationships between the data in appendix and the variables/parameters in the model were improved.

A pseudocode for the genetic algorithm implementation was added.

The objective/fitness function was added with detailed description of each variable, followed by all constraints that describe the feasible region of the problem.

Each variable/parameter in the constraints was described as well.

Line 223 (now Line 173): As suggested, a correction was made regarding the relation between the model, solution procedure and the genetic algorithm.

Line 229 (now Line 201) and Line 232 (now Line 215): The relation (equation) used to calculate the costs in terms of general risks was added with the respective explanation – Line 209.

The costs in terms of ergonomic risks were already explained. They were based on the assessment of ergonomic risks related to the expenditure of metabolic energy that occurs during the execution of the various tasks within each work area, and they were obtained by estimation based on comparison with several tasks already studied by [33,34,35] having been made an average for each work zone. An example was given by an additional table (table 1) for a better understanding. An additional explanation is given later after Line 297.

Line 244 (now Line 234): Symbols for variables and parameters were used in the equations as suggested.

Figure 6: The figure was removed and the information was transferred to text in a generic way.

Line 282 (now Line 291): The variables and indices were written in italic, according to the SI standard, and an explanation was made about the calculation of each variable and the respective units.

Line 343 (now Lines 304 and 489): An explanation of the role of the Euclidian distance in the model was made. This distance is included in the objective function affecting it directly.

Line 413 (now Line 479): Title changed to “Optimization results” as suggested.

Line 421 (now Line 493): An explanation of the origin of the results from Table 4 was given. The results were obtained from: the general risk assessment that was performed by applying the William T. Fine Method (Appendix A) (scenarios 1 and 3) and the ergonomic risk assessment that was performed through the metabolic energy expenditure estimation (Appendix B) (scenarios 2 and 4).

Lines 423–526 (Now Lines 499 – 631): An explanation about the difference between the size of the sections on Figure 7 (now Figure 3) and the sections on Figures 8 and 9 (now Figures 4 and 5) was given (Lines 505 - 509)

“Once the optimization model was applied with the aim of reorganizing the productive areas in order to mitigate occupational risks, allying, if possible, with the operative efficiency, the results are shown in a simplified way, as uniform squares distributed along a rectangular layout, without relating directly to the various stages of production with different sizes shown in the winery layout example”

Points of interest

The representation in on Figures 8 and 9 this way it is to provide a simplicity of the representation. The constraints of the different production areas had been accounted for the model as described in the Lines 282 – 369.

One of the main purposes of this work was to retrofit an existing winery. Thus, an additional figure was added (Figure 6) in order to show if the results were feasibly providing some ideas for future work.

Reviewer 2 Report

This paper addresses the use of Meta-Heuristic Model for Optimization of Production Layouts Based on Occupational Risk Assessment. The overall article represents a very interesting application of the genetic algorithm for reducing ergonomic risks. Despite the merits, there are some issues to be considered. I recommend processing Figure 1 into a black and white version without effects. In your case, you put together Introduction and Material and Methods. Try to separate this into the two parts so that the Introduction contains background, hypotheses, similar research to the field, their comparison, and how your research differs from it. The end of the Introduction would include a clear definition of what the article is dealing in its core. What tools did you use and the brief result of your study. Subsequently, the Material and Methods section would contain a description of the tools and methods, i.e. theoretical basis and your proposal for a solution. In the Results section, you only describe the results of your proposed solution with the results in a clear form. In my opinion, your Results are from the current Chapter 4 and partly 3. You can break Table 1 into bullets to save space, you currently have 36 pages. Part of Conclusion should be renamed Discussion as it corresponds to this section in content. Conclusion would be appropriate if part of Disscusion was too long, which is not your case. Overall is article good and I wish the authors good luck in further research.

Author Response

Thank you for all your comments and suggestions.

1. Figure 1 was changed into a black and white version without effects.

2. As suggested, the paper was reorganized into four sections:

    • Introduction: this section contains the background, hypotheses, similar research on the field, their comparison, and how our research differs from it. At the end of this section, it was included a clear definition of what the article is dealing in its core, what methods were used and a brief result of our study was given.
    • Material and Methods: this part contains a description of the method, including a theoretical basis and our proposal for a solution.
    • Optimization results: This section describes the results of our proposed solution. The results were reorganized to provide a better understanding.
    • Discussion: The title conclusion was changed to Discussion as suggested. The information was reorganized.

Observation: Table 1 was removed. A new part related to similar research on the field was inserted providing a better understanding.

Round 2

Reviewer 1 Report

/